# Factors associated with continuation of hormonal contraceptives among married women of reproductive age in Gilgit, Pakistan: a community-based case–control study

Fazila Bibi [1], Sarah Saleem [2], Shiyam S Tikmani,[2] Shafquat Rozi[2]

¹Center of Excellence for Trauma and Emergencies, Aga Khan University, Karachi, Pakistan
²Community Health Sciences, Aga Khan University, Karachi, Pakistan

**Correspondence to**
Fazila Bibi;
fazila.sahibjan@aku.edu

## ABSTRACT

**Objective** This study aimed to determine the factors associated with continuation of hormonal contraceptive methods among married women of Gilgit, Pakistan at least 6 months after their initiation.

**Design** Unmatched case–control study.

**Setting** Community settings of Gilgit, Pakistan from 1 April 2021 to 30 July 2021.

**Participants** The cases were married women of reproductive age who, at the time of interview, were using a hormonal method of contraception for at least 6 months continuously, and controls were married women of reproductive age who had used a hormonal method in the past and currently were using a non-hormonal method for at least 6 months.

**Primary and secondary outcome measures** OR for continuation of hormonal contraceptive.

**Results** The factors significantly associated with continuous use of hormonal contraceptive methods for our sample from Gilgit were the family planning centre's distance from home (adjusted OR (AOR) 6.33, 95% CI 3.74 to 10.71), satisfaction with current method used (AOR 3.64, 95% CI 2.06 to 6.44), visits to the family planning centre to avail services (AOR 1.86, 95% CI 1.07 to 3.45) and relatively older age of women (AOR 1.07, 95% CI 1.02 to 1.12). In addition, women with formal education (AOR 0.27, 95% CI 0.12 to 0.6) were less likely to use a modern contraceptive method.

**Conclusion** Continuation of using a hormonal method was associated with easy access to family planning centres, satisfaction with the current method and frequent visits to the family planning centres. Continuation of using a hormonal method was also seen in women with low education status. The importance of the presence of family planning centres near residential areas cannot be emphasised more. This does not only provide easy access to family planning methods, but also reassure women of continuation of modern methods when they face any unpleasant effects while using these.

## INTRODUCTION

Access to safe and effective contraceptives is the right of every woman and man regardless of their religion, socioeconomic status,

## STRENGTHS AND LIMITATIONS OF THIS STUDY

⇒ The case-to-control ratio was kept at 1:2, which enhances the likelihood that our findings are representative of the experiences of the general population.

⇒ Subjective assessment of the factor 'walking distance' due to a lack of internet access in all union councils of Gilgit.

⇒ This study is generalisable to areas that are geographically similar to Gilgit.

⇒ Data collection was done during the COVID-19 pandemic; thus, women might have discontinued going to family planning centres for hormonal contraceptives due to fear of getting infection, and this might have affected our results.

region or country. This ensures that individuals can make well-informed decisions about the contraceptive method that best suits their preferences and needs.[1] The trends of contraceptive use, its discontinuation and switching to a different method are significant identifiers of how well family planning programmes are fulfilling the needs of family planning of women of reproductive age and couples in a country.[2] Worldwide, 922 million women of reproductive age use any form of contraceptives[3]; however, there are still 12% of the women who want to avoid pregnancy but are not using any method of contraception.[4] High fertility rates and low contraceptive prevalence rates (CPRs) are the main elements of exponential population growth in South Asian countries.[5] In Pakistan, the combination of a high fertility rate and low contraceptive utilisation plays a significant role in contributing to unfavourable reproductive health indicators, particularly concerning maternal mortality. Furthermore, discontinuation of contraception has been found to be one of the major reasons of induced abortions in Pakistan. Multiple Indicator Cluster

Survey of Pakistan (2016–2017) reported that in Gilgit-Baltistan (GB), 38.1% of married women of reproductive age had access to contraception, of these 32.1% use one of the modern contraceptive methods.[6]

Contraceptive continuation is the process of uninterrupted usage of contraceptive methods for at least 6 months. Hormonal methods use hormones to regulate ovulation and prevent pregnancy, and include injectable birth controls, contraceptive skin patch, vaginal ring and contraceptive pills. Factors associated with contraceptive continuation are efficacy of the method, frequency of doses, and client-related characteristics, such as age, education level, parity, etc.[7 8] Distance of the family planning centre from home and availability of different methods at the centre are additional important factors contributing to continuity of using a method. Presence of healthcare staff at the centre, level of education, age, number of living children and area of residence were all found to be associated with contraceptive continuation.[9–12]

There is notable scarcity of information regarding predictors of sustained use of hormonal contraceptives in Pakistan, and information concerning the continuation of any contraceptive method is generally lacking. Findings from this study could play a pivotal role in enhancing CPR and decreasing the incidence of discontinuation of hormonal methods, which are more effective when compared with non-hormonal method alternatives. Additionally, promoting factors linked to the continued use of hormonal methods can aid in preventing unintended pregnancies, thereby contributing to improved maternal health outcomes.

GB is situated in the northern part of Pakistan, which covers an area of over 72 971 km$^2$ and is highly hilly and mountainous. Gilgit is the capital and largest district with an estimated population of 216 760.[13] Seventy-nine per cent of family planning centres with trained staff in Gilgit are in the main town, while the remaining are in other union councils with 4–29% trained staff[13] as shown in figure 1.

This suggests that all specialised facilities are centred in Gilgit town and women from other union councils must travel to get specialised services. However, the average distance from Gilgit town to the nearest health facility is 2.7 km, which is less far than other union councils that are 18–24 km away from the nearest health facility.[13] Considering the high efficacy of the hormonal methods to prevent pregnancy, we were interested in identifying region-specific (Gilgit) factors that were associated with the continuation of hormonal contraceptives.

## METHODS

We used a community-based unmatched case–control study design, with a case-to-control ratio of 1:2. Cases were married women of reproductive age who were currently using a hormonal method of contraception for at least 6 months, and controls were married women of reproductive age who had ever used a hormonal method but are now using a non-hormonal method for at least 6 months. This was to know why women switched from a hormonal to a non-hormonal method, and to make groups comparable, we selected the control group with a history of hormonal method use. We collected data from the catchment population of female heath workers of three union councils—Gilgit town, Danyore and Nomal. Hormonal method users were sexually active women who were using one of the hormonal contraceptive methods, that is, pills, injectables or implants for at least 6 months without disruption at the time of enrolment in the study, and controls were sexually active married women of reproductive age, with a history of using a hormonal method but stopped it before 6 months after initiating it and switched to a non-hormonal contraceptive method for at least 6 months. We excluded women using intrauterine contraceptive

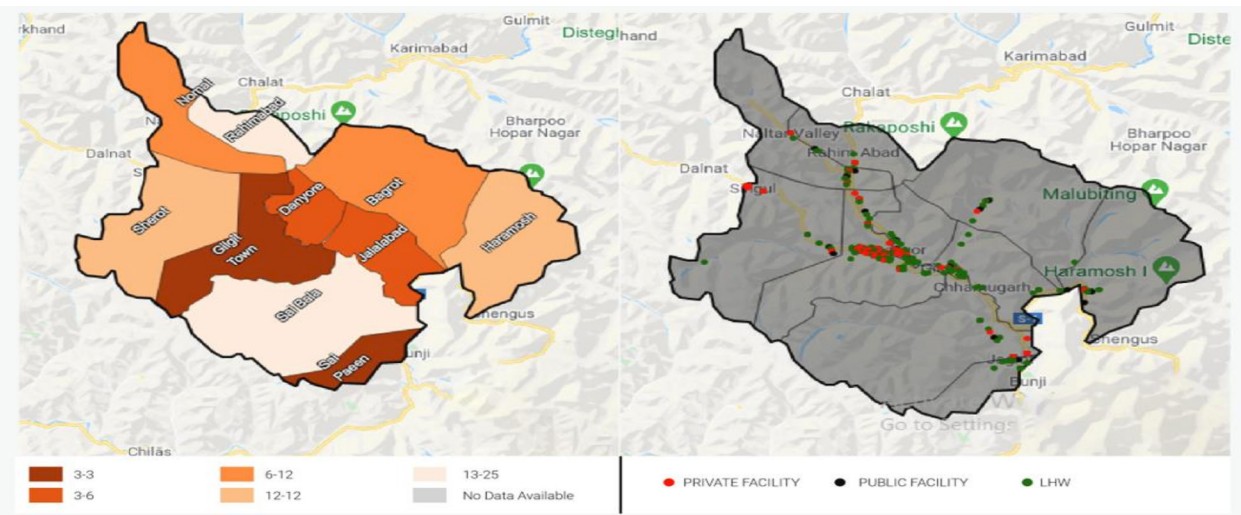

**Figure 1** Left side: map of Gilgit. Right side: average distance of public and private facilities in Gilgit. Red dots show private facilities, black dots show public facilities and dark green shows female health workers (LHWs, lady health worker).

device (IUCD), as there are of two types (hormonal and non-hormonal) of methods and they might not know the type of IUCD they were using. Women who were using a permanent method of contraception were also excluded. Data were collected from March 2021 to July 2021.

To account for a non-response rate, the sample size was increased by 20%; to determine an OR of at least 2, by taking power of 80% and significance of 5% with the ratio of 1:2 between cases and controls, a minimum of 141 cases and 282 controls were required for this study.[4]

Data were analysed using Stata V.14. Frequency and percentages are reported for qualitative variables. Mean and SD are reported for continuous quantitative variables. We used logistic regression for both univariate (simple logistic regression) and multivariable analyses to identify factors associated with continued use of a hormonal contraceptive method. Multicollinearity among all independent variables significant at univariate level was checked considering 0.80 as a cut-off for Pearson correlation (for continuous variables), and Cramer's V to check multicollinearity among categorical variables.

### Patient and public involvement

There is no patient and public involvement in the current study.

## RESULTS

### Description of study population

We completed interviews on 141 cases and 282 controls. Table 1 illustrates the sociodemographic characteristics of women who continued using hormonal methods for at least 6 months and current non-hormonal method users who used hormonal methods and discontinued them. The mean age of the women who were using hormonal

**Table 1** Sociodemographic characteristics of women who continued and discontinued using hormonal contraceptives

| Variables | Women who continued using hormonal contraceptives, n=141 n (%) | Women who dicontinued using hormonal contraceptives, n=282 n (%) |
|---|---|---|
| Age of women (years) | | |
| Mean (SD) | 34.8 (5.9) | 31.5 (5.3) |
| Area of residence | | |
| Rural | 45 (31.9) | 64 (22.7) |
| Urban | 96 (68.0) | 218 (77.3) |
| Education | | |
| No formal education | 30 (21.2) | 18 (6.3) |
| Formal education | 111 (78.7) | 264 (93.6) |
| Language | | |
| Brushaski | 96 (68.0) | 204 (72.3) |
| Others | 45 (31.9) | 78 (27.6) |

**Table 2** Frequency distribution of biological factors among women who continued and discontinued using hormonal contraceptives

| Variables | Women who continued using hormonal contraceptives, n=141 n (%) | Women who dicontinued using hormonal contraceptives, n=282 n (%) |
|---|---|---|
| Age at initiating contraceptive use | | |
| Mean (SD) | 26.2 (3.8) | 25.4 (3.1) |
| Number of living children at the time of initiating contraceptive use | | |
| Mean (SD) | 2 (1.4) | 1 (1.2) |
| Decision-maker of contraceptive use | | |
| Self | 18 (12.7) | 22 (7.8) |
| Mainly husband | 33 (23.4) | 136 (48.2) |
| Joint decision | 90 (63.8) | 124 (43.9) |
| Source of current method | | |
| Government | 114 (81.4) | 90 (78.9) |
| Private | 26 (18.5) | 24 (21.0) |
| Information given regarding side-effects | | |
| Yes | 124 (87.9) | 260 (87.9) |
| No | 17 (12.0) | 22 (7.8) |
| Information given regarding other methods | | |
| Yes | **98 (69.5)** | **214 (75.8)** |
| No | **43 (30.5)** | **68 (24.1)** |
| Experienced side-effects | | |
| Yes | 119 (84.4) | 7 (2.4) |
| No | 22 (15.6) | 275 (97.5) |
| Satisfied with current method | | |
| Yes | 118 (83.6) | 168 (59.5) |
| No | 23 (16.3) | 114 (40.4) |
| Want more effective method | | |
| Yes | 125 (44.3) | 35 (24.8) |
| No | 157 (55.6) | 106 (75.1) |
| Information given by female health worker | | |
| Mother and child health | 6 (4.2) | 65 (23.0) |
| Contraceptive supplies, family planning and family health counselling | 120 (85.1) | 209 (74.1) |
| Others | 15 (10.6) | 8 (2.8) |

Bold values indicate that 69.5% of the continued users were given information regarding other available methods while 75.8% of discontinued users were given information regarding availability of other than chosen method.

methods was 34.8±5.9 years, whereas the mean age of those who discontinued using them was 31.5±5.3 years.

Table 2 presents the frequency distribution of biological factors among women who continued using hormonal contraceptives and those who discontinued using them. Among those who continued using them, at the time of interview, 24.1% were using pills, 2.1% were using implants and 73.4% were using injectables for contraception. Conversely, among those who discontinued using hormonal contraceptives, nearly 59.0% were using traditional methods and 40.7% were using condoms at the time of the interview. Before transitioning to non-hormonal methods, these women were using pills (62.0%), injectables (37.5%) and implants (0.35%). The mean age at initiating contraceptive use among those who continued using hormonal contraceptives was 26.2 years (SD 3.8) and among those who discontinued using them was 25.4 years (SD 3.1). Notably, there were some differences observed between the two groups in terms of number of living children they had at the time of initiating contraceptive use. Women who continued using hormonal contraceptives had a mean of two children (SD 1.4), while those who discontinued using them had one child (SD 1.2).

## Factors related to service delivery

Table 3 provides an overview of factors related to service delivery among women who continued and discontinued using hormonal contraceptives. All the women in our sample were aware of the nearest family planning centres in their area. Around 84.4% of women who continued using and 73.0% of women who discontinued using hormonal contraceptives had been to family planning centres to avail services in the last 6 months. The distance from home to the nearest family planning centre was measured subjectively by asking women if they could walk to access the nearest centre or require transport to reach to the centre. Approximately 79.0% of the women who continued uing hormonal methods mentioned that the family planning centre is within walking distance, while 21.2% needed transport to reach to the nearest family planning centre. In contrast, 38.3% of the women who discontinued using hormonal contraceptives were able to reach the centre on foot, while 61% needed transport.

In both the groups, those women who did not visit family planning clinics mentioned unsuitable timings of the family planning centre, far distance, non-availability of family planning staff, cost of services, unavailability of the desired method and transportation issues as some of the reasons for not availing the services.

Women were asked about their satisfaction with the services provided at home by the female health workers. It was found that 65.2% of women among those who continued using and 77.3% women among those who discontinued using hormonal contraceptives expressed satisfaction with the home services they received.

On multivariable analysis, after adjusting for effect of other variables in the model, walking distance to a family

**Table 3** Frequency distribution of factors related to service delivery among women who continued and discontinued using hormonal contraceptives

| Variables | Women who continued using hormonal contraceptives, n=141 n (%) | Women who dicontinued using hormonal contraceptives, n=282 n (%) |
|---|---|---|
| Ever been to FP centres to avail services | | |
| Yes | 119 (84.4) | 205 (72.7) |
| No | 22 (15.6) | 77 (27.3) |
| Reasons for not going to FP centres | | |
| Facility-related issues | 4 (18.1) | 27 (35.0) |
| Service-related issues | 13 (59.0) | 34 (44.1) |
| Transport-related issues | 5 (22.7) | 16 (20.7) |
| Number of days of services provided by centre | | |
| Mean (SD) | 6.0 (1) | 5.7 (0.9) |
| Routine fee charged by facility | | |
| Yes | 133 (94.3) | 277 (98.2) |
| No | 8 (5.6) | 5 (1.7) |
| Presence of healthcare staff at centre | | |
| Yes | 105 (74.4) | 163 (57.8) |
| No | 36 (25.5) | 119 (42.2) |
| Counselling services at FP centre | | |
| Yes | 75 (53.1) | 133 (47.1) |
| No | 66 (46.8) | 149 (52.8) |
| Satisfied with services provided at home | | |
| Yes | 92 (65.2) | 218 (77.3) |
| No | 49 (34.7) | 64 (22.7) |
| Reasons for dissatisfaction | | |
| Did not like method | 12 (24.4) | 10 (15.6) |
| Others (have more side-effects, provide counselling only) | 37 (75.5) | 54 (84.3) |

FP, family planning.

planning centre (adjusted OR (AOR)=6.33, 95% CI=3.74 to 10.71), visiting a family planning centre (AOR=1.86, 95% CI=1.07 to 3.45), older age of the women (AOR=1.07, 95% CI=1.02 to 1.12) and having formal education (AOR 0.27, 95% CI=0.13 to 0.47) were independently associated with continuation of hormonal contraceptives as shown in table 4.

## DISCUSSION

The findings of our study highlight the importance of both service delivery and individual factors in influencing the continuation of hormonal contraceptive use for a minimum of 6 months among women in GB. Among the service delivery-related factors, significant contributors included the proximity of the family planning centre to

**Table 4** Adjusted OR (95% CI) for the factors associated with continuation of hormonal contraceptives among women who continued and discontinued using hormonal contraceptives in Gilgit, Pakistan

| Variables | Women who continued using hormonal contraceptives, n=141 n (%) | Women who dicontinued using hormonal contraceptives, n=298 n (%) | Adjusted OR (95% CI) |
|---|---|---|---|
| Distance from home to FP centre | | | |
| Need transport | 30 (21.2) | 174 (61.7) | 1 |
| Walking distance | 111 (78.7) | 108 (38.3) | 6.4 (3.70 to 11.19) |
| Age of women (years) | | | |
| Mean (SD) | 34.8 (5.9) | 31.5 (5.3) | 1.06 (1.01 to 1.11) |
| Education | | | |
| No formal education | 30 (21.2) | 18 (6.3) | 1 |
| Formal education | 111 (78.7) | 264 (93.6) | 0.27 (0.13 to 0.47) |
| Satisfied with current method | | | |
| No | 23 (16.3) | 114 (40.4) | 1 |
| Yes | 118 (83.6) | 168 (59.5) | 3.64 (2.06 to 6.44) |
| Visits FP centre | | | |
| No | 22 (15.6) | 77 (27.3) | 1 |
| Yes | 119 (84.4) | 205 (72.7) | 1.86 (1.07 to 3.45) |

Only statistically significant variables are shown in table 4.
FP, family planning.

the women's residential area, the quality of information provided by the family planning centre staff regarding method side-effects, overall satisfaction with the current contraceptive method and regular visits to the nearest family planning centre. Additionally, noteworthy individual factors that played a significant role were age and education status of the women.

Our results are in concordance with other studies.[14] Proximity to the health facilities and facilities providing a wide range of contraceptives are related to increased use of contraceptive methods. Physical access to family planning services and other maternal and neonatal health facilities are mostly determined by travel distance.[15–18] Access to family planning centres within 1.9 km and family centres that provide four or more methods are also the factors that increase contraceptive use. However, as distance increases from 2 km irrespective of the number of methods provided by the facility, the contraceptive use starts to decline.[19] Previous studies from Pakistan also found that proximity to functional family planning centres is strongly associated with contraceptive use.[16 18] Our study also suggests that walking distance from home to family planning centres matters for those who use a contraceptive method.

In our study sample, women who had ever been to formal school were less likely to use a hormonal contraceptive compared with those who had never been to formal school. Conventionally, women with no formal education are less likely to practise modern contraceptive use compared with women with some formal education[20]; thus, women's education was a strong predictor of

contraceptive use and women who had attained secondary or more education were twice likely to avail contraceptive services.[21] However, there can be religious reasons for preferring non-invasive and non-chemical methods. Upperclass women face more facilitating conditions for traditional contraceptive use such as better husband–wife communication, which is so important for effective birth control. More importantly, they may be better able to deal with contraceptive failure because they have the capacity to afford an additional child and they also have easy access to abortion.[22] We believe, based on our study, that closer distance to family planning centres and easy access to contraceptive method of choice have made education a less important factor for women with regard to using a contraceptive method. In the context of Gilgit, living near family planning centres and door-to-door visits by female health workers have made women in the region more aware about the contraceptive methods irrespective of education status.

Our study found that satisfaction with the method currently used is one of the significant reasons to continue hormonal methods of contraception. Clients' satisfaction with a method makes users compliant to use it and becomes a main factor for its continuation. In our study, all the women said that they have been visited by a female health worker in the last 6 months and they had received contraceptive counselling (side-effects, how to manage side-effects and seek treatment in case of experiencing side-effects) and referral. Being visited by a female health worker to discuss family planning in the last 12 months is positively associated with continuing

long-acting reversible contraceptives.[23] Similarly, women in Malawi who had been visited by female health workers were more likely to continue using injectables compared with other contraceptives.[19] Hence, female health workers are critical sources of information for introducing women to hormonal contraceptives.[24] Female health workers in Gilgit provide door-to-door contraceptive counselling and make women aware of possible side-effects of a contraceptive method and refer them and counsel them about ways to manage and seek treatment for the side-effects. We found positive association of contraceptive use with age. Women aged 20–44 years prefer short-acting compared with women aged 15–19 years who prefer long-acting contraceptives.[25–27] As the family of a woman becomes complete, she desires to limit pregnancies, and hence contraceptive compliance increases as the age of the woman increases.[25 26]

Our study possessed several notable strengths. First, we employed a robust recruitment process based on female health workers' records, who routinely update and maintain accurate records during their frequent visits. Additionally, we corroborated this information with the women themselves before enrolling them in the study. This method significantly reduced the likelihood of misclassifying the outcome status of the participants.

Furthermore, we maintained a balanced case-to-control ratio of 1:2, which enhances the likelihood that our findings are representative of the experiences of the general population. We also employed a consistent data collection tool for both cases and controls, ensuring the reliability and comparability of the data.

However, it is important to acknowledge the primary limitation of our study, which was the subjective assessment of the distance of family planning centres. This limitation arose from the lack of internet access in all union councils of Gilgit. Nevertheless, women who never used transport to reach family planning centres were supportive of giving an affirmation of the proximity of family planning centres from their homes. Additionally, our data collection occurred during the COVID-19 pandemic, a period during which women may have refrained from visiting family planning centres due to concerns about contracting the virus. This factor could potentially have had an impact on our results. Furthermore, this study is generalisable to areas that are geographically similar to Gilgit.

## CONCLUSION AND RECOMMENDATIONS

In Gilgit, continuation of using hormonal methods was notably linked with easy access to family planning centres, satisfaction with current method used and visits to family planning centres. Our study findings recommend that family planning centres should be closer to the residential areas to make them easily accessible to women. Moreover, appropriate counselling, reassurance and response to side-effects are key for continuation of using hormonal methods. Counselling regarding managing side-effects

by female health workers at home and by healthcare providers in the centres should be ensured. Looking ahead, conducting a cohort study to observe behaviour of women for using, continuing, switching and discontinuing hormonal methods could provide valuable insights into the factors influencing the continuous use of these methods.

### Dissemination plan

The findings of this study will be presented at important national and international conferences and published in peer-reviewed scientific journals.

**Contributors** FB—conceptualisation, data curation, data analysis, investigation, methodology, software and writing original draft. SS—conceptualisation, methodology, data curation, supervision and writing (review and editing). SST—methodology and writing (review and editing). SR—data analysis and editing. FB and SS are overall content guarantors.

**Funding** FB received funding from HRP Alliance, part of UNDP-UNFPA-UNICEF-WHO-World Bank Special Programme of Research, Development and Research Training in Human Reproduction (HRP), a co-sponsored programme executed by the WHO, to complete her studies.

**Disclaimer** The author is a staff member of the World Health Organization. The author alone is responsible for the views expressed in this publication and they do not necessarily represent the views, decisions or policies of the World Health Organization. This article represents the views of the named authors only and does not represent the views of the WHO. The main funder (HRP) had no role in study design, data collection and analysis.

**Map disclaimer** The inclusion of any map (including the depiction of any boundaries therein), or of any geographic or locational reference, does not imply the expression of any opinion whatsoever on the part of BMJ concerning the legal status of any country, territory, jurisdiction or area or of its authorities. Any such expression remains solely that of the relevant source and is not endorsed by BMJ. Maps are provided without any warranty of any kind, either express or implied.

**Competing interests** None declared.

**Patient and public involvement** Patients and/or the public were not involved in the design, or conduct, or reporting, or dissemination plans of this research.

**Patient consent for publication** Not required.

**Ethics approval** This study involves human participants and the ethical review committee of Aga Khan University gave ethical approval for conducting the study (2022-6023-21033). Written informed consent was obtained from all eligible women who agreed to participate in the study.

**Provenance and peer review** Not commissioned; externally peer reviewed.

**Data availability statement** The data is available from coresponding author upon reasonable request.

**ORCID iDs**
Fazila Bibi http://orcid.org/0000-0003-3011-3631
Sarah Saleem http://orcid.org/0000-0002-6797-8631

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
