## [Reviewer comments · BMJ Open]

ARTICLE DETAILS

TITLE (PROVISIONAL)	Factors associated with continuation of hormonal contraceptives among married women of reproductive age in Gilgit- Pakistan: a community-based case control study
AUTHORS	Bibi, Fazila; Saleem, Sarah; Tikmani, Shiyam; Rozi, Shafquat

VERSION 1 – REVIEW

REVIEWER	Mondal, Sourav International Institute for Population Sciences, Fertility and Social Demography
REVIEW RETURNED	12-Jun-2023

GENERAL COMMENTS	First of all I would like to congratulate the authors for this well written manuscript. This manuscript could be improved more by incorporating few changes which I am suggesting as follows: 1. Firstly, In Introduction section, the definition of hormonal contraceptives and which are the methods that comes under the hormonal contraceptive needs to be written.2. The data had been collected during Covid19 times, there might be any effect of that in the study results? If so, it can be mentioned in the limitation of the study.3. "Multicollinearity among all independent variables significant at univariate level was checked considering 0.80 as cut off" - using which method multicollinearity had been checked? please mention that.4. In the Table 4, had there only the statistically significant results been shown? If this then a Because there is no demarcation about significance like using "*" or any other symbols.5. The name of the software is "Stata" not " STATA", please correct it.6. For giving more plausible explanation of "women who had ever been to formal school were less likely to use a hormonal contraceptive compared to those who had never been to formal school" you can take help of this article (Alaka Malwade Basu (2005) Ultramodern contraception, Asian Population Studies, 1:3, 303-323, DOI: 10.1080/17441730500441178). All the best in making the changes. Thank you
--

REVIEWER	Mihretie, Gedefaye Debre Tabor University, Midwifery
REVIEW RETURNED	07-Jul-2023

GENERAL COMMENTS	Dear Editors, I would like to thank you for considering me to review the manuscript entitled 'Factors associated with continuation of hormonal contraceptives among married women of reproductive age in Gilgit Pakistan: a community-based case control study' General comment Thank you, dear authors, I entirely reviewed your manuscript, and it was well written. But I have some issues. What do the authors expect to gain from their work? There have been many studies about the discontinuation of hormonal contraceptives. What is your special focus within your specific area of interest? Please briefly describe why did you select the controls that had a history of hormonal contraception before? The controls should be married women with no history of hormonal contraceptives, unless control selection is doubtful. What does simple regression mean? THANK YOU
---

VERSION 1 – AUTHOR RESPONSE

1. Hormonal methods that use hormones to regulate ovulation and prevent pregnancy that include injectable birth controls, contraceptive skin patch, vaginal ring and oral contraceptive pills.
 2. • Data was collection was done during Covid19 times; women might have discontinued going to family planning centers due to fear of getting infection and this might have affected our results.
 3. Multicollinearity among all independent variables significant at univariate level was checked considering 0.80 as cut off for Pearson correlation (for continuous variables), Crammrs V to check multicollinearity among categorical variables.
 4. symbol "*" has been used to indicate statistically significant variables in Table 4
 5. name of software corrected (Stata)
 6. However, there can be religious reasons for preferring non-invasive and non-chemical methods. Upper class women face more facilitating conditions for traditional contraceptive use such as better husband wife communication, which is so important for effective birth control and more importantly, they may be better able to deal with contraceptive failure because they have capacity to afford an additional child and they also have easy access to abortion (27).
- Reviewer 2
- controls were married women of reproductive age who had ever used a hormonal method but now are using a non-hormonal method at least for last six months because we wanted to know why women switched from hormonal to non-hormonal method and to make groups comparable we selected control group with history of hormonal method use.
- . Findings from this study would help in improving contraceptive prevalence rate and decreasing discontinuation of hormonal methods which are more effective compared to non-hormonal method. Moreover, promoting factors that are associated with the continuation of hormonal method will help to avoid untimed and unwanted pregnancies hence it will improve maternal health.

VERSION 2 – REVIEW

REVIEWER	Mondal, Sourav International Institute for Population Sciences, Fertility and Social Demography
REVIEW RETURNED	18-Sep-2023
GENERAL COMMENTS	All suggestions has been incorporated. This paper can be accepted in its current form. All the best for authors.